# Randomized controlled trial of the effectiveness of olive and black seed oil combination on pain intensity and episiotomy wound healing in primiparous women: A study protocol

**Romina Fili**[1], **Fereshteh Behmanesh**[2]*, **Sana Nazmi**[1], **Maryam Nikpour**[3], **Zahra Memariani**[4]

**1** Babol University of Medical Sciences, Babol, I.R. Iran, **2** Social Determinants of Health Research Center, Health Research Institute, Babol University of Medical Sciences, Babol, I.R. Iran, **3** Non-Communicable Pediatric Diseases Research Center, Health Research Institute, Babol University of Medical Sciences, Babol, I.R. Iran, **4** Pharmaceutical Sciences Research Center, Babol University of Medical Sciences, Babol, I.R. Iran

* f.behmanesh2015@gmail.com

**Funding:** This study was financially supported by Babol University of Medical Sciences. Award Number: 724134467. F.B will receive the fund.URL:

## Abstract

### Background

Episiotomy is associated with side effects, such as pain and wound infection. Additionally, discomfort after episiotomy affects the quality of life of both the mother and the baby. Medicinal herbs are one alternative method for the treatment of episiotomy wounds. This study will investigate the effectiveness of the combination of olive and black seed oil on pain intensity and the healing of episiotomy wounds in primiparous women.

### Methods

This randomized clinical trial will be conducted on primiparous women who have had a normal delivery with an episiotomy. There are 3 groups in this study: one group will receive a combination of olive oil and black seed oil, another group will receive olive oil alone, and the use of oils will start 24 hours after delivery. Ten drops will be applied topically 3 times a day for 10 days. The third group (control) will receive only routine care. Data will be collected through a demographic characteristics questionnaire, REEDA (Redness, Edema, Ecchymosis, Discharge, and Approximation) Scale, and Visual Analog Scale. To determine and compare the effects of pharmaceutical interventions on pain intensity and episiotomy wound healing in the groups, an analysis of variance (ANOVA) test with repeated measurements will be used with SPSS version 22.

### Discussion

The results of this study will show the effects of a combination of olive and black seed oil, as well as olive oil alone, on pain intensity and episiotomy wound healing in primiparous

https://www.mubabol.ac.ir/ This funder contributed to the approval of the study. There was no additional external funding received for this study.

**Competing interests:** The authors have declared that no competing interests exist.

women. The positive effects observed in this trial with these oils could be valuable for women who have undergone an episiotomy.

## Introduction

Episiotomy is an incision in the area between the perineum and the anus, performed to enlarge the perineum and facilitate the process of natural childbirth, especially in primiparous women [1–3]. The prevalence of episiotomy varies in different countries and races [4]. For example, in Asian countries, due to the anatomical features of the perineum, episiotomy is used as a method to facilitate natural delivery. But there is no inferable statistics of the prevalence of episiotomy in Iran [5]. However, the rate of using episiotomy in a study in Iran was 41.5% [6]. This intervention is associated with side effects such as pain [3, 7, 8]. Studies have shown that approximately 30% of women experience pain in the first two weeks, and 7% experience it up to 3 months after natural delivery [9]. Due to the lack of direct observation of the mother and the proximity of the wound area to the anus, there is a possibility of wound infection [5, 10, 11]. Discomfort after episiotomy affects the quality of life of both the mother and the baby [3, 12, 13]. Considering that the baby requires the mother's presence and support for growth and development, especially in the early days after birth, it is crucial to pay attention and provide support to mothers who have undergone an episiotomy [10, 14].

Since Iran's current policy is to increase the birth rate and population, more attention should be paid to prenatal and post-pregnancy care, especially for vaginal childbirth. An unpleasant experience during childbirth can affect a mother's future pregnancies and may lead to her deciding not to have more children, ultimately reducing population growth [9]. On the other hand, if episiotomy is not used, there is a risk of perineal tears that may cause problems for the mother, such as a reduced quality of sexual life [7, 15] and difficulties with breastfeeding [2, 16]. In Iran, episiotomy is still common, with a prevalence of over 41% among primiparous women [17].

To alleviate the pain intensity of episiotomy and facilitate its faster healing, various treatment methods, including oral, rectal, and sitz baths, are used [14, 18]. Given the increasing prevalence of herbal medicine in recent decades [19] and the adverse effects associated with chemical drugs, such as high costs [7] and mild drowsiness to cardiotoxicity [20], it is advisable to consider herbal medicine, following the recommendations of the World Health Organization [21]. It appears that plant oils, such as olive oil and black seed oil, may be effective in promoting the healing and repair of perineal wounds [21, 22]. According to studies, many medicinal plants, especially olive oil, can accelerate wound healing and reduce pain and complications caused by episiotomy [16, 23]. In a previous study by the author, the use of olive oil sitz baths was found to be effective in reducing pain intensity and promoting episiotomy wound healing in primiparous women [14]. Olive oil is rich in substances that provide it with antimicrobial, antifungal, and anti-inflammatory properties [24, 25]. These include phenolic compounds that promote cell repair, antioxidants that help reduce inflammation, and oleocanthal that can alleviate pain [14, 23, 26]. Moreover, various studies have shown the beneficial effects of olive oil [27], its unsaturated fatty acids, including oleic acid [28], and its phenolic compounds [29] on wound healing. Olive oil has been found effective in treating skin diseases, suppressing the growth of viruses and bacteria, and promoting wound healing [14, 30]. Black seed contains 30% to 40% oil and is particularly rich in a type of antioxidant called thymoquinone [31, 32], as well as phenols, linoleic acid, and oleic acid, which have been the subject of various studies demonstrating their positive effects on wound healing [33]. Additionally, black seed exhibits anti-inflammatory and analgesic properties, along with antiviral and antifungal

effects [34, 35]. These effects can be attributed to thymoquinone, which is one of the main constituents of black seed [36]. Thymoquinone is a monoterpene diketone with a molecular weight of 164.204 g/mol and is considered a lipophilic compound [37]. In fact, most of the plant's properties are attributed to thymoquinone [38]. Studies have shown that black seed is suitable for promoting wound healing in the human body and can help prevent infections [35, 39]. Furthermore, adding black seed oil to olive oil enhances its antioxidant properties [40]. Given the limited number of human clinical studies on black seed oil and the scarcity of research regarding the local effectiveness of the combination of olive oil and black seed oil in alleviating pain and promoting episiotomy wound healing, the present study aims to evaluate the effectiveness of the combination of olive and black seed oil in reducing pain intensity and facilitating episiotomy wound healing in primiparous women. This study will be implemented in Shahid Yahya Nejad Hospital, Babol city. In this hospital, low risk mothers are admitted for delivery. Routine care after episiotomy in these mothers includes sitting in a basin of warm water, and the mothers do not use any oil to repair episiotomy. Recently, some gynecologists prescribe Nivasha spray for faster healing of episiotomy wounds, but these mothers will not be included in the study due to interference with the results of our study. This study adopts a parallel design to compare the side effects caused by drug use, such as burning, itching, episiotomy wound opening, and the need for repair, among 3 groups.

## Materials and methods

### Study design

This randomized clinical trial will be conducted as a parallel-group, non-inferiority study on postpartum women who have undergone episiotomy in the hospitals of Babol city. The schedule of enrolment, interventions, and assessments was shown in Fig 1. The research design is based on the interpretive standards of trial reports (Fig 2). This study was approved by the Ethics Committee of Babol University of Medical Sciences (code: IR.MUBABOL.REC.1401.145). Additionally, the study protocol was registered in the Iranian Registry of Clinical Trials (code: IRCT20180218038783N2) (S1 File). All methods will be performed in accordance with the relevant guidelines and regulations, including the Declaration of Helsinki, and written informed consent will be obtained from all participants. The trial results will be communicated to participants, health care professionals, the public, and other relevant groups, as well as researchers and sponsors, through the publication of results and reporting in results databases. The SPIRIT Checklist has been shown in S1 Checklist.

### Eligibility criteria

Women interested in participating in the study will be invited. After explaining the research objectives and obtaining written informed consent, the researcher will proceed with convenient sampling. Then, based on the inclusion criteria, the samples will be randomly divided into two intervention groups and one routine care group.

Inclusion criteria are primiparous women, natural vaginal delivery with episiotomy, literacy, and BMI < 30. Exclusion criteria are unwillingness of participants to continue participating in the study, fourth-degree tear, history of dermatitis, history of allergy and eczema diagnosed related to plants, and history of gestational diabetes and overt diabetes.

### Study sample and sampling

The required sample size was estimated to be 36 based on a previous study on pain intensity of episiotomy using G Power software. The effect size (mean difference) was determined to be

| | STUDY PERIOD | | | | | | | |
|---|---|---|---|---|---|---|---|---|
| | **Enrolment** | **Allocation** | **Post-allocation** | | | | | **Close-out** |
| **TIMEPOINT\*\*** | $T_1$ | *0* | *$10^{th}$ day* | | | | | 2024 December |
| **ENROLMENT:** | | | | | | | | |
| **Eligibility screen** | X | | | | | | | |
| **Informed consent** | X | | | | | | | |
| **Allocation** | X | | | | | | | |
| **INTERVENTIONS:** | X | | | | | | | |
| *[Olive Oil]* | X | X | ←→ | | | | | |
| *[olive oil plus black seed oil]* | X | X | ←→ | | | | | |
| *[Control]* | X | X | ←→ | | | | | |
| **ASSESSMENTS:** | | | | | | | | |
| *[intensity of pain in the episiotomy area and healing of episiotomy wound]* | X | | X | | | | | X |
| *[side effects: (burning and itching, episiotomy wound opening and need for repair, The need for painkillers)]* | X | | X | | | | | X |
| *[Demographic variables]* | | X | | | | | | X |

**Fig 1. The schedule of enrolment, interventions, and assessments.**

1.4 (pain intensity of episiotomy between the intervention and control groups), with an α of 0.05 and a β of 5% [20].

The researcher will select the samples according to the inclusion criteria and randomly allocate them into 3 groups of 36 individuals using a random-numbers table. The block size will be 6, and the table will be managed by a researcher who is not involved in the sampling process. The person responsible for sampling (a midwifery student) will receive the intervention code from the managing researcher after selecting the sample based on the inclusion criteria. To blind the random allocation list, a unique code will be assigned to each of the intervention groups, known only to the herbal pharmacist associate. The supervisor, researcher, and statistical analyst will be blinded to ensure proper management. To ensure the reliability of this

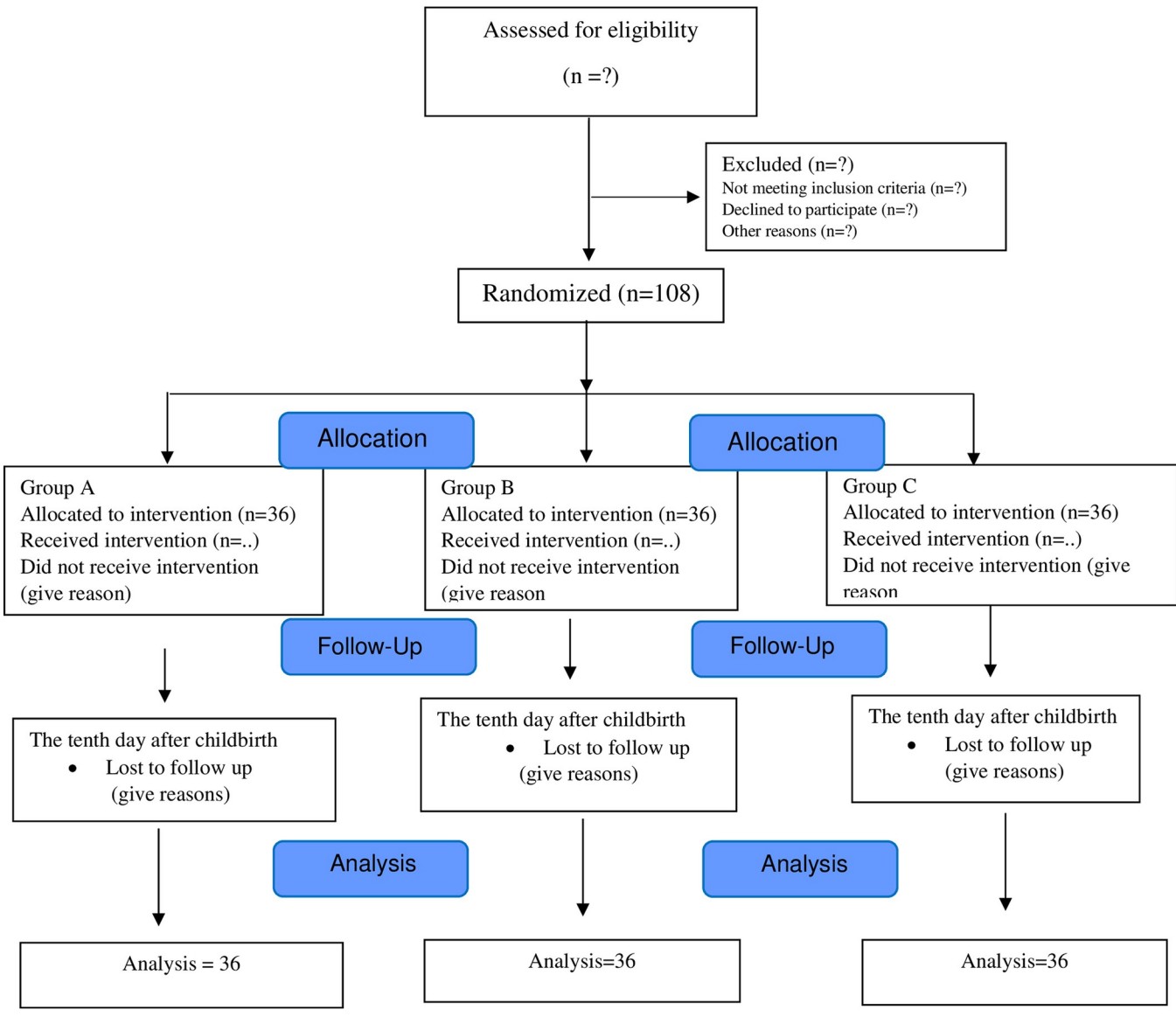

**Fig 2. CONSORT (Consolidated Standards of Reporting Trials) flow diagram.**< /Figure_Caption>

research, the person who examines the episiotomy wound on the first and tenth days will be the same for all mothers. A skilled and trained midwifery student will collect the data by examining the mothers.

## Data collection and management

This research will be conducted after obtaining permission from the Ethics Committee of Babol University of Medical Sciences, registering with the Iranian Registry of Clinical Trials, and obtaining the necessary sampling permits. After contacting the midwifery department of the hospitals affiliated with Babol University of Medical Sciences, the researcher invites eligible women to participate in the study. The researcher will first explain the study's purpose and the nature of the intervention to the women. The researcher ensures the confidentiality of their information, clarifies her responsibility for any treatment complications, and assures them

that the treatment interventions will not change if they choose not to participate. Participants are then requested to provide written consent. Additionally, participants are assured that they can withdraw from the study at any stage. Subsequently, the researcher (a midwifery student) informs the supervisor about the sample's inclusion in the study, and the supervisor shares the intervention code with the researcher.

All study subjects will be educated on following post-delivery health tips and caring for episiotomy wounds. The intervention groups (one group receiving olive oil plus black seed oil and the other receiving olive oil alone) will also receive instructions on how to use the containers containing the oils. Women in the intervention groups will use the oils 24 hours after delivery, applying 10 drops topically 3 times a day for 10 days. Given that each milliliter contains 15 drops, and mothers are required to use the oils 3 times a day, approximately 30 mL of each type of oil will be provided to mothers for the duration of 10 days.

The participants will be instructed to wash and dry their perineum before applying the oils. They should then wear gloves and slowly apply 10 drops of oil to the perineum, massaging it in with their hands. The control group will receive only routine care. All mothers will be provided with painkillers as needed. The type and number of painkillers used by each group will be recorded using a checklist completed by the mother over 10 days. This data will be evaluated and considered as an auxiliary variable in the final analysis. All mothers will be examined again on the 10th day after delivery to determine the severity of pain and healing of the episiotomy wound.

In this research, 3 questionnaires will be administered within the first 24 hours and 10 days after delivery.

1. Demographic Characteristics Questionnaire: This questionnaire comprises 22 questions that gather information such as age, education level, place of residence, employment status, height, weight, number of pregnancies and abortions, duration of hospitalization during labor, receipt of antibiotics during labor and post-delivery, and hospitalization of the baby in the intensive care unit. The questionnaire was designed based on the research objectives to identify potential confounding variables.

2. Visual Analog Scale for Measuring Episiotomy Pain Intensity: The Visual Analog Scale, also known as the McGill Pain Ruler, is a linear scale used for pain assessment. It consists of a 10-cm ruler, with 0 indicating no pain and 10 indicating the most severe pain possible. Between these 2 endpoints, the scale is divided into 3 levels: mild pain (1–3), moderate pain (4–7), and severe pain (8–10) [41]. Perineal pain intensity refers to the pain experienced by mothers in their perineal area and is self-reported before and after the intervention.

3. The Redness, Edema, Ecchymosis, Discharge, and Approximation Scale: The REEDA (Redness, Edema, Ecchymosis, Discharge, and Approximation) Scale is a standardized tool for assessing episiotomy wound healing, including variables such as swelling, bruising, discharge, redness, and the distance between the 2 edges of the episiotomy wound. Each of these variables is scored on a scale from 0 to 3. The researcher records the scores for each variable separately after observing and examining the patient both before and after the examination. The overall score on the scale ranges from 0 to 15, with higher scores indicating less favorable wound healing and lower scores indicating better wound healing. In a reliability analysis conducted by Alvarenga et al. (2015), the kappa coefficient was reported for this scale. The results indicated good agreement for the variable of redness ($0.46 <$ Kappa $\geq 0.66$), marginal to good agreement in the first 3 assessments of edema ($0.16 <$ Kappa $\geq 0.46$), marginal agreement in the evaluation of ecchymosis ($0.25 <$ Kappa $\geq 0.42$), and good agreement in the assessment of the discharge item ($0.75 <$ Kappa $\geq 0.88$) [42].

The primary outcomes of this study include the assessment of pain intensity in the episiotomy area and the evaluation of episiotomy wound healing. Secondary outcomes include the monitoring of symptoms such as burning and itching, as well as the occurrence of episiotomy opening and the need for painkillers. These secondary outcomes will be assessed through a combination of examinations conducted by the researcher and self-reports provided by the mothers. Additionally, any side effects reported by each participant will be carefully reviewed, and the total number of side effects within each group will be documented.

The product used in the study is a combination of cold-pressed black seed oil and standard extra-virgin olive oil, both of which adhere to national standards. The standard extra-virgin olive oil is manufactured by a pharmaceutical-grade company called Pishgaman Chemi, while the cold-pressed black seed oil is obtained from the pharmaceutical company Barij Essance. These two oils are mixed in equal proportions [43]. Importantly, both products meet approved specifications in terms of microbial and pollution control, as well as chemical and physical standards. The process of mixing and packaging these two oils is conducted under hygienic conditions within the traditional pharmacy laboratory of the Faculty of Traditional Medicine.

## Data analysis

The analysis method used in this study is intention-to-treat analysis, ensuring the maintenance of the random allocation process.

All analyses will be conducted using SPSS version 22, including both descriptive and analytical analyses. A comprehensive outline of the analysis procedures will be documented in the statistical analysis plan.

Descriptive information will be presented using means and SDs or as numbers and percentages. To assess the correlation between quantitative variables, Pearson's correlation test will be used, while the chi-square test will be used to examine the correlation between 2 qualitative variables.

For comparing pain intensity and episiotomy wound healing after the intervention in the 3 groups, the analysis of covariance (ANCOVA) test will be used, taking into account the measurements taken prior to the intervention. Multilevel linear regression will be conducted to adjust for confounding variables, including demographic, midwifery, and delivery-related factors.

Additionally, the effect size will be demonstrated through mean differences and 95% CIs. A significance level of $P < 0.05$ will be considered as the threshold for statistical significance.

## Report

Researchers will enhance their interaction with individuals to ensure comprehensive data collection. To achieve this, we will establish more robust communication channels with the participants via social media and phone to provide medication reminders.

In this process, one of the researchers (RF) will assume responsibility for data collection, while another individual (SN) will be tasked with entering the data into the SPSS software. A data manager (MN) will review all the data without knowledge of the treatment allocations. These procedures will be conducted under the supervision of FB.

## Discussion

This study aims to investigate the effectiveness of a combination of olive and black seed oil on pain intensity and episiotomy wound healing in primiparous women. Olive oil is known for its rich content of substances that impart antimicrobial, antifungal, and anti-inflammatory properties [24, 25], including peniphenol that promotes cell repair, antioxidants that help

reduce inflammation, and oleocanthal that suppresses pain [14, 23, 26]. Black seed oil, on the other hand, contains 30% to 40% oil and is particularly rich in thymoquinone, a type of antioxidant [31, 32], along with phenols, linoleic acid, and oleic acid, which various studies have shown to be effective in wound healing [33]. Studies have shown that black seed oil is suitable for wound healing in the human body and can prevent infection [35, 39]. Moreover, combining black seed oil with olive oil enhances its antioxidant properties [40].

Skin wound healing is a critical physiological process that involves the collaboration of various cell types and their products. The repair process begins early in the inflammatory phase, leading to the replacement of specialized structures through collagen deposition and regeneration, which involves cell proliferation and subsequent differentiation from existing tissue cells [44].

While the individual effects of black seed oil [21] and olive oil [9, 14] on episiotomy have been evaluated in previous studies, this research conducted by the study team is the first to examine the combined impact of these 2 oils on episiotomy. Clinical research on whether topical application of this oil combination can effectively promote episiotomy repair and alleviate perineal pain severity is limited. If the results of this study show that the combination of these 2 substances is indeed effective in reducing episiotomy pain intensity and enhancing healing, it could be proposed as a cost-effective treatment option. One of the major limitations of the current study is that some mothers may not return on the 10th day due to the recovery or because of hospitalization of the baby and another limitation is that sterile tape won't be used and visual inspection will be used in the assessing of wound healing.

## Conclusion

The results of this study will show the effects of the combination of olive oil and black seed and olive oil alone on pain intensity and episiotomy wound healing in primiparous women. The potential positive outcomes from testing these oils may propose a viable solution for the use of low-risk methods after childbirth.

## Supporting information

**S1 Checklist. SPIRIT 2013 checklist.**
(PDF)

**S1 File.**
(PDF)

## Acknowledgments

The authors would like to thank the Deputy of Research and Technology of Babol University of Medical Sciences and mothers who will participate in this study.

## Author Contributions

**Conceptualization:** Romina Fili, Fereshteh Behmanesh, Sana Nazmi, Maryam Nikpour, Zahra Memariani.

**Data curation:** Fereshteh Behmanesh.

**Formal analysis:** Maryam Nikpour.

**Investigation:** Romina Fili, Fereshteh Behmanesh.

**Methodology:** Romina Fili, Fereshteh Behmanesh, Maryam Nikpour, Zahra Memariani.

**Project administration:** Fereshteh Behmanesh.

**Resources:** Fereshteh Behmanesh.

**Software:** Romina Fili, Fereshteh Behmanesh, Zahra Memariani.

**Supervision:** Fereshteh Behmanesh.

**Validation:** Fereshteh Behmanesh.

**Visualization:** Romina Fili, Fereshteh Behmanesh.

**Writing – original draft:** Romina Fili, Fereshteh Behmanesh.

**Writing – review & editing:** Romina Fili, Fereshteh Behmanesh, Sana Nazmi, Maryam Nik-pour, Zahra Memariani.

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
