## [Editor Report · Decision Letter 0]

30 Nov 2023

PONE-D-23-34640Randomized controlled trial of the effectiveness of olive and black seed oil combination in pain intensity and episiotomy wound healing in primiparous women: study protocolPLOS ONE

Dear Dr. Fereshteh Behmanesh,

Thank you for submitting your manuscript to PLOS ONE. After careful consideration, we feel that it has merit but does not fully meet PLOS ONE’s publication criteria as it currently stands. Therefore, we invite you to submit a revised version of the manuscript that addresses the points raised during the review process.Improve the general outlook of the protocol by restructuring sentences, make your statement clear and logical to enable your readers understand your work.  Also, attend to incomplete sentences to use of words.Make protocol succinct and avoid repetitions. The "Material and methods" section can be structured to reflect the following:Study designEligibility criteriaStudy sample and samplingData collection and managementData analysisReportAuthors can then resubmit the protocol for review.

We look forward to receiving your revised manuscript.

Kind regards,

Eunice Bolanle Turawa, MSc

Academic Editor

PLOS ONE

Journal Requirements:

"This study was financially supported by Babol University of Medical Sciences. Award Number: 724134467. F.B will receive the fund.

URL: https://www.mubabol.ac.ir/

This funder contributed to the approval of the study."

"The authors would like to thank the Deputy of Research and Technology of Babol University of Medical Sciences for supporting the project."

"This study was financially supported by Babol University of Medical Sciences. Award Number: 724134467. F.B will receive the fund.

URL: https://www.mubabol.ac.ir/

This funder contributed to the approval of the study."

5. Please note that in order to use the direct billing option the corresponding author must be affiliated with the chosen institute. Please either amend your manuscript to change the affiliation or corresponding author, or email us at plosone@plos.org with a request to remove this option.

7. We note that the original protocol that you have uploaded as a Supporting Information file contains an institutional logo. As this logo is likely copyrighted, we ask that you please remove it from this file and upload an updated version upon resubmission.

Additional Editor Comments:

Study protocol." This is a clinical trial involving first-time mothers in early postpartum period. The protocol requires general restructuring for clarity, ease communication flow, and scientific soundness. Incomplete and unclear sentences should be improved on. Also, make the protocol succinct without repetitions.

---

## [Author Response · Author response to Decision Letter 0]

7 Dec 2023

Dear editor

PLOS ONE

Thank you and the honorable Reviewer (s) for your valuable comments for the manuscript entitled: Randomized controlled trial of the effectiveness of olive and black seed oil combination on pain intensity and episiotomy wound healing in primiparous women: study protocol (PONE-D-23-34640). These comments are very useful in the scientific improvement of the manuscript. Responses to the following comments are given:

1. Improve the general outlook of the protocol by restructuring sentences, make your statement clear and logical to enable your readers understand your work. Also, attend to incomplete sentences to use of words.

Regards, corrected.

2. Make protocol succinct and avoid repetitions. 

These items have been corrected and marked in the text with track changes.

3. The "Material and methods" section can be structured to reflect the following:

Study design

Eligibility criteria

Study sample and sampling

Data collection and management

Data analysis

Report

Sincerely, corrected. 

Done

Done

Done

Corrected.

Journal Requirements:

This manuscript meets PLOS ONE's style requirements, including those for file naming.

"This study was financially supported by Babol University of Medical Sciences. Award Number: 724134467. F.B will receive the fund.

URL: https://www.mubabol.ac.ir/

This funder contributed to the approval of the study."

Corrected

"The authors would like to thank the Deputy of Research and Technology of Babol University of Medical Sciences for supporting the project."

"This study was financially supported by Babol University of Medical Sciences. Award Number: 724134467. F.B will receive the fund.

URL: https://www.mubabol.ac.ir/

This funder contributed to the approval of the study."

The Acknowledgments Section corrected: The authors would like to thank the Deputy of Research and Technology of Babol University of Medical Sciences and mothers who will participate in this study.

The amended statements was included in the cover letter.

This is a protocol study and data availability are not applicable. This sentence is included in the manuscript and cover letter.

5. Please note that in order to use the direct billing option the corresponding author must be affiliated with the chosen institute. Please either amend your manuscript to change the affiliation or corresponding author, or email us at plosone@plos.org with a request to remove this option.

I emailed to plosone@plos.org with a request to remove the direct billing option.

Corrected. It deleted from declarations.

7. We note that the original protocol that you have uploaded as a Supporting Information file contains an institutional logo. As this logo is likely copyrighted, we ask that you please remove it from this file and upload an updated version upon resubmission.

I removed it from the original protocol and uploaded an updated version upon resubmission.

Corrected based on Supporting Information guidelines.

---

## [Decision Letter · Decision Letter 1]

4 Jan 2024

PONE-D-23-34640R1.

Randomized controlled trial of the effectiveness of olive and black seed oil combination on pain intensity and episiotomy wound healing in primiparous women: Study protocol.

Dear Dr. Behmanesh,

Thank you for submitting your manuscript to PLOS ONE. After careful consideration, we feel that it has merit but does not fully meet PLOS ONE’s publication criteria as it currently stands. Therefore, we invite you to submit a revised version of the manuscript that addresses the points raised during the review process. Be sure the entire manuscript undergoes thorough editing, focusing on grammar, spelling, and typo errors. It is advisable you use English expert services. We anticipate your revised manuscript and appreciate your attention to these details.

We look forward to receiving your revised manuscript.

Kind regards,

Eunice Bolanle Turawa, MSc

Academic Editor

PLOS ONE

Reviewers' comments:

Reviewer's Responses to Questions

**Comments to the Author**

1. Does the manuscript provide a valid rationale for the proposed study, with clearly identified and justified research questions?

Reviewer #1: Yes

Reviewer #2: No

Reviewer #3: Yes

2. Is the protocol technically sound and planned in a manner that will lead to a meaningful outcome and allow testing the stated hypotheses?

Reviewer #1: Yes

Reviewer #2: Yes

Reviewer #3: Yes

3. Is the methodology feasible and described in sufficient detail to allow the work to be replicable?

Reviewer #1: Yes

Reviewer #2: No

Reviewer #3: Yes

4. Have the authors described where all data underlying the findings will be made available when the study is complete?

Reviewer #1: Yes

Reviewer #2: No

Reviewer #3: No

5. Is the manuscript presented in an intelligible fashion and written in standard English?

Reviewer #1: No

Reviewer #2: No

Reviewer #3: Yes

6. Review Comments to the Author

You may also provide optional suggestions and comments to authors that they might find helpful in planning their study.

Reviewer #1: Dear Author

The manuscript has valuable results but needs essential edition as below:

1- English writing is poor. There are a lot of spelling and grammar errors and editing by a native speaker is necessary.

2- In the introduction:

• The mention that not performing an episiotomy leads to severe tearing is not true in all mothers and depends on the case.

• In reference number 6, the sentence is incomplete.

• Start sentence with “such as” is unusual.

3- In the methods section:

• First, the type of study and then the work method should come.

• Objectives should be removed from the beginning of the methods section.

• In reliability and validity of Reeda tool, please wright the number of reliability and validity.

• In inclusion criteria: the sentence is incomplete.

4- Discussion:

• Discussion need to an essential revision.

• According to the results, conclusion need revision.

References:

• Some references should be changed basis on guideline format.

• Revise the reference list according to Paper Submission Guide: Abstract the name of journals.

• More references should be covering the last 5 years.

Be succesful

Reviewer #2: Thank you as you invited me to review this manuscript. Please see my comments as follows:

First of all this study has some grammatical and typo errors and should be revised by an expert person in English literature.

Abstract:

1. Method: While one group will receive olive oil plus black seed and one group will receive olive oil alone, why the third group do not receive placebo, and they will receive routine care?

Introduction

1. The second point of "highlight" should be revised, as the main reason for using herbs is their safety and not the high costs of pharmaceutical treatment.

2. Authors stated that if not using the episiotomy, there is a perineal tears that may cause problem for mother. Non-indicated episiotomy causes many problems for mother such as sexual dysfunction and pain. Please mention the rate of episiotomy especially non-indicated in Iran.

3. Please be careful and exact when you are talking about side effects of medications. After delivery, women just receive an analgesic for reducing their pain and analgesics are not expensive and also do not have many side effects.

4. Please use the results of those studies that used olive oil for reducing pain intensity after episiotomy in the introduction.

Methods

1. Women who do not use the oil regularly is not exclusion criteria, but they should consider drop-out.

2. Please move the objectives of the study to the end of introduction.

3. Some of exclusion criteria are drop-out that happen during study such as "women who do not use olive oil regularly"

4. Authors stated that they excluded fourth degree tears. Is that mean you will recruit second and third degree tears?

5. Please provide more details about intervention, e.g. what should participants do before applying the oil on the perineum, any washing?

6. Please write how much oil will be given to each participant?

7. The midwife or a gynecologist that perform episiotomy and type of repair are important factors that did not mention by authors in this study.

8. Authors should advise participants to take the similar pain killers.

9. A significance level of 0.05 will be considered in all tests. It should be <0.05.

Reviewer #3: This is an interesting study looking at the effect of olive oil and black seed on improving on pain intensity and

episiotomy wound healing in primiparous women. This is an inferiority study.

Some comments where the details reported could be clarified further.

1. As this a non-inferiority study, the sample size is missing some fundamental information, i.e what was defined as the clinical minimal difference that would constitute non-inferior. Type 1 error information should be included in sample size calculation. In addition as this is a 3 arms trial, is the non-inferiority hypothesis in relation to control vs EACH intervention, make this comparison explicit and define this.

2. For the randomisation process, please indicate the allocation ratio. No need to state the "The size of the blocks is 6, and 18 blocks of 6 are produced to create a sequence of size 108". You could just state, variable block size. And perharps include this here " (One group, olive oil plus black seed oil and another group of olive oil alone) and control group is routine care"

3. Unsure what this sentence means "The supervisor, researcher and statistical analyst will be blinded to perform proper management." Do you mean blinded analysis?

4. Define primary outcome, i.e pain intensity- how is this measured? How will healing be measured?

5. Same comment for Secondary outcomes, how will these be defined?

6. Define what you mean by intention to treat analysis, i,e people analysed according to the group there were assigned regardless of taking allocated intervention.

7. More information in the analysis section, i.e mention that data will be reported to CONSORT guidelines as this an RCT.

8. In the analysis section as this an RCT, no need to carry out formal tests on baseline characteristics as this not recommended, since any differences would occur by chance. Also mention that all details regarding analysis will be stated in the statistical analysis plan.

9. What is justification of using multi-level modelling? I.e what are the repeated measures of random effect variable?

10. How will adherence/compliance be assessed?

7. PLOS authors have the option to publish the peer review history of their article (what does this mean?). If published, this will include your full peer review and any attached files.

Reviewer #1: No

Reviewer #2: **Yes: **Parvin Abedi

Reviewer #3: No

---

## [Author Response · Author response to Decision Letter 1]

15 Jan 2024

Dear Editor in Chief

PONE-ONE

Thank you for your thorough review and consideration of our manuscript “Randomized controlled trial of the effectiveness of olive and black seed oil combination on pain intensity and episiotomy wound healing in primiparous women: Study protocol" (PONE-D-23-34640R1) that was submitted to the PONE-ONE. 

We believe that the comments and suggestions that were recommended by the reviewers have informed a much improved and more fully developed paper that will offer an important contribution to the field. We have highlighted the changes that were undertaken in response to your comments in the revised manuscript. Responses to each of your comments are below. 

Thanks for your kind attention to the manuscript.

Sincerely, Corresponding author

Dr Fereshteh Behmanesh

Reproductive Health

 

Reviewers Response

Reviewer 1

The manuscript has valuable results but needs essential edition as below:

1- English writing is poor. There are a lot of spelling and grammar errors and editing by a native speaker is necessary. Thank you very much for the opinion of the respected reviewer, according to the suggestion of dear reviewer, this manuscript will be edited by a native and the certificate will be sent for Journal.

2- In the introduction:

• The mention that not performing an episiotomy leads to severe tearing is not true in all mothers and depends on the case. Corrected, p 3, par 1

In reference number 6, the sentence is incomplete. Corrected, p 3, par 2

Start sentence with “such as” is unusual. Corrected, p 4, line 1 & p10, par 3

In the methods section:

• First, the type of study and then the work method should come. First, the type of study and then the work method writhed. P 5, par 2

Objectives should be removed from the beginning of the methods section. Removed and the third objective added in the end of introduction. P 5, par 1

• In reliability and validity of Reeda tool, please wright the number of reliability and validity. Corrected. P 8 & 9

In inclusion criteria: the sentence is incomplete. Corrected. P6, par 2

Discussion:

• Discussion needs to an essential revision. In the protocol study, considering that the findings are not clear, the discussion cannot be written in detail. However, it was corrected according to the comment of the respected referee.

According to the results, conclusion need revision. Corrected. P11, par 2

References:

• Some references should be changed basis on guideline format.

• Revise the reference list according to Paper Submission Guide: Abstract the name of journals.

• More references should be covering the last 5 years. Corrected based on guideline format.

Reviewer 2

First of all this study has some grammatical and typo errors and should be revised by an expert person in English literature. Thank you very much for the opinion of the respected reviewer, according to the suggestion of dear reviewer, this manuscript will be edited by a native and the certificate will be sent for Journal.

Abstract:

1. Method: While one group will receive olive oil plus black seed and one group will receive olive oil alone, why the third group do not receive placebo, and they will receive routine care? This is a very good question. By the way, the research team had also thought about this issue. But since the use of any oil may have a positive or negative effect on wound healing, for this reason placebo was not used in the control group.

Introduction 1. The second point of "highlight" should be revised, as the main reason for using herbs is their safety and not the high costs of pharmaceutical treatment. With respect to the reviewer’s comment, corrected. P 3

2. Authors stated that if not using the episiotomy, there is a perineal tears that may cause problem for mother. Non-indicated episiotomy causes many problems for mother such as sexual dysfunction and pain. Please mention the rate of episiotomy especially non-indicated in Iran. With respect to the reviewer's comment, unfortunately, the incidence of non-indicated episiotomy in Iran was not found in our search. Perhaps because if an episiotomy is non-indicated, the doctor or midwife will not report it due to legal claims.

 The following sentence was used instead: In Iran, episiotomy is still common and its prevalence is reported in more than 41% of primiparous women.

3. Please be careful and exact when you are talking about side effects of medications. After delivery, women just receive an analgesic for reducing their pain and analgesics are not expensive and also do not have many side effects. Thanks to the reviewer, this sentence was removed. Of course, in Iran, in addition to painkillers, most doctors routinely prescribe antibiotics after delivery with episiotomy.

4. Please use the results of those studies that used olive oil for reducing pain intensity after episiotomy in the introduction. Added. P4, par 2

Methods

1. Women who do not use the oil regularly is not exclusion criteria, but they should consider drop-out. While thanking the respected reviewer, since our analysis method is ITT, the women should consider drop-out based on reviewer comment. This exclusion criteria were written incorrectly in the manuscript and has now been removed. P 6

2. Please move the objectives of the study to the end of introduction. Removed and the third objective added in the end of introduction. P 5, par 1

3. Some of exclusion criteria are drop-out that happen during study such as "women who do not use olive oil regularly" While thanking the respected reviewer, since our analysis method is ITT, the women should consider drop-out based on reviewer comment. This exclusion criteria were written incorrectly in the manuscript and has now been removed. P 6

4. Authors stated that they excluded fourth degree tears. Is that mean you will recruit second and third degree tears? Yes, that is right. We will recruit second- and third-degree tears.

5. Please provide more details about intervention, e.g. what should participants do before applying the oil on the perineum, any washing? Provided. The samples were asked to wash and dry their perineum before using oils. Then wear gloves and slowly pour 10 drops of oil on the perineum and massage with hands. P 8, par 1

6. Please write how much oil will be given to each participant? Considering that each cc contains 15 drops and mothers should use oils three times a day, approximately 30 cc of each type of oil was given to mothers for 10 days. P 8, par 1

7. The midwife or a gynecologist that perform episiotomy and type of repair are important factors that did not mention by authors in this study. The type of repair will be evaluated by questionnaire but the midwife or a gynecologist that perform episiotomy will not be checked. However, in the author's previous study, this issue was investigated and had no effect on the results of the study.

8. Authors should advise participants to take the similar pain killers The type of painkiller and its number per day in three groups is evaluated by a checklist that is completed by the mother within ten days and then evaluated and considered as auxiliary variable in the final analysis.

9. A significance level of 0.05 will be considered in all tests. It should be <0.05. corrected

Reviewer 3

This is an interesting study looking at the effect of olive oil and black seed on improving on pain intensity and episiotomy wound healing in primiparous women. Some comments where the details reported could be clarified further.

1. As this a non-inferiority study, the sample size is missing some fundamental information, i.e what was defined as the clinical minimal difference that would constitute non-inferior. Type 1 error information should be included in sample size calculation. In addition as this is a 3 arms trial, is the non-inferiority hypothesis in relation to control vs EACH intervention, make this comparison explicit and define this.

Thank you very much for the comments of the respected reviewer

According to the suggestion of dear reviewer. Sample size corrected

2. For the randomisation process, please indicate the allocation ratio. No need to state the "The size of the blocks is 6, and 18 blocks of 6 are produced to create a sequence of size 108". You could just state, variable block size. And perharps include this here " (One group, olive oil plus black seed oil and another group of olive oil alone) and control group is routine care" According to the suggestion of dear reviewer, corrected

3. Unsure what this sentence means "The supervisor, researcher and statistical analyst will be blinded to perform proper management." Do you mean blinded analysis? As in this sentence: the statistical analyst does not know about the study groups and it will be determined after the statistical analysis (only the pharmacist knows)

4. Define primary outcome, i.e pain intensity- how is this measured? How will healing be measured? The measurement of pain intensity and wound healing is written in detail in the subtitle of "data collection and management" of the method section in paragraphs 5 and 6.

According to the suggestion of dear reviewer, definition of pain intensity and wound healing was added. 

The intensity of perineal pain is the pain that mothers feel in their perineal area and it is recorded by self-report method before and after the intervention.

Skin wound healing is an essential physiological process that consists of the cooperation of many cell strains and their products. Attempts to repair the lesion caused by local invasion begin very early in the inflammatory phase. Finally, they lead to repair, which involves the replacement of specialized structures caused by collagen deposition and regeneration, which is related to the process of cell proliferation and posterior differentiation through cells already present in tissue cells.

5. Same comment for Secondary outcomes, how will these be defined? The primary outcomes are pain intensity in the episiotomy area and healing of episiotomy wound. The secondary outcomes are Burning and itching and episiotomy opening and the need for painkillers which will be evaluated through examination by the researcher and through mother's self-report.

6. Define what you mean by intention to treat analysis, i,e people analysed according to the group there were assigned regardless of taking allocated intervention. We will use intention-to-treat analysis in this study, because intention-to-treat analysis will provide an unbiased estimate of the efficacy of the intervention at the level of adherence in the study.

7. More information in the analysis section, i.e mention that data will be reported to CONSORT guidelines as this an RCT.

 Statistical analysis according to CONSORT guidelines was written. 

8. In the analysis section as this an RCT, no need to carry out formal tests on baseline characteristics as this not recommended, since any differences would occur by chance. Also mention that all details regarding analysis will be stated in the statistical analysis plan. According to the suggestion of dear reviewer, this sentence was removed

9. What is justification of using multi-level modelling? I.e what are the repeated measures of random effect variable? Multilevel linear regression will be done for adjust confounders variables (demographic, midwifery and delivery). In the section of statistical analysis was added and repeat measurement analysis was removed. 

10. How will adherence/compliance be assessed? Multilevel linear regression will be done for adjust confounders variables (demographic, midwifery and delivery).

---

## [Decision Letter · Decision Letter 2]

23 Feb 2024

PONE-D-23-34640R2Randomized Controlled Trial of the Effectiveness of Olive and Black Seed Oil Combination on Pain Intensity and Episiotomy Wound Healing in Primiparous Women: A Study ProtocolPLOS ONE

Dear Dr. Behmanesh,

Thank you for submitting your manuscript to PLOS ONE. After careful consideration, we feel that it has merit but does not fully meet PLOS ONE’s publication criteria as it currently stands. Therefore, we invite you to submit a revised version of the manuscript that addresses the points raised during the review process.

**ACADEMIC EDITOR: Please respond to all reviewers comments**

We look forward to receiving your revised manuscript.

Kind regards,

Ahmed Mohamed Maged, MD

Academic Editor

PLOS ONE

Reviewers' comments:

Reviewer's Responses to Questions

**Comments to the Author**

1. Does the manuscript provide a valid rationale for the proposed study, with clearly identified and justified research questions?

Reviewer #3: Yes

Reviewer #4: Yes

Reviewer #5: Yes

2. Is the protocol technically sound and planned in a manner that will lead to a meaningful outcome and allow testing the stated hypotheses?

Reviewer #3: Yes

Reviewer #4: Partly

Reviewer #5: Yes

3. Is the methodology feasible and described in sufficient detail to allow the work to be replicable?

Reviewer #3: Yes

Reviewer #4: No

Reviewer #5: Yes

4. Have the authors described where all data underlying the findings will be made available when the study is complete?

Reviewer #3: Yes

Reviewer #4: No

Reviewer #5: Yes

5. Is the manuscript presented in an intelligible fashion and written in standard English?

Reviewer #3: Yes

Reviewer #4: Yes

Reviewer #5: Yes

6. Review Comments to the Author

You may also provide optional suggestions and comments to authors that they might find helpful in planning their study.

Reviewer #3: All comments have been addressed.

Reviewer #4: Dears

• Report rate of episiotomy in Iran?

• In the introduction: Explain setting of the study and how used type of oil therapy people (primiparous) based on culture.

• There are many confounding variables which must be controlled. In Iranian hospitals, the most childbirth is performed by gynecological assistants or midwifery students, with many interventions; confounding variables: for example, time and type of Epi repair, the number of cutgut chromic sutures, Length and depth of episiotomy. fetal information: fetal weight, fetus position of the occiput posterior or anterior, labor information: duration of the first and second stage, removal of the placenta by hand and etc.

• Isn’t report the type of random allocation process.

• How will adherence/compliance be assessed?

• How many times a day were used the oils?

• Was it used based on individual demand and then will be recorded.

• Isn’t report Reliability this research.

• From the ethical point of view, all three groups should have received painkiller, it should be reported which group used less.

• Report of side effect essential oils?

• when Intensity pain and wound healing will be measured?

• How Intensity pain and wound healing will be measured? in REEDA scale for evaluate items; sterile tapes should be used to mark the amount of bruising and redness and... then measure with a ruler.

• Limitation? For example, this primiparous they change our location during first days, Especially the tenth day, from the mother's house to her house and they are not present at the previous address.

Sincerely

Reviewer #5: Thank you very much for giving me the opportunity to review this manuscript.

I congratulate the authors for the work done.

I don't have comment.

7. PLOS authors have the option to publish the peer review history of their article (what does this mean?). If published, this will include your full peer review and any attached files.

Reviewer #3: No

Reviewer #4: No

Reviewer #5: No

---

## [Author Response · Author response to Decision Letter 2]

6 Mar 2024

Dear Editor 

PONE-ONE

Thank you for your thorough review and consideration of our manuscript “Randomized controlled trial of the effectiveness of olive and black seed oil combination on pain intensity and episiotomy wound healing in primiparous women: Study protocol" (PONE-D-23-34640R1) that was submitted to the PONE-ONE. 

We believe that the comments and suggestions that were recommended by the reviewers have informed a much improved and more fully developed paper that will offer an important contribution to the field. We have highlighted the changes that were undertaken in response to your comments in the revised manuscript. Responses to each of your comments are below:

Thanks for your kind attention to the manuscript.

Sincerely, Corresponding author

Dr Fereshteh Behmanesh

Reproductive Health

 

Reviewers Response

Reviewer #3: 

All comments have been addressed. Thank you very much for your valuable comments in the previous review.

Reviewer #4 

1-Report rate of episiotomy in Iran? Thank you very much for the attention of the honorable reviewer, rate of episiotomy in Iran added to the introduction. (P4, Par 2)

2- In the introduction: Explain setting of the study and how used type of oil therapy people (primiparous) based on culture.

 Corrected (P 6)

There are many confounding variables which must be controlled. In Iranian hospitals, the most childbirth is performed by gynecological assistants or midwifery students, with many interventions; confounding variables: for example, time and type of Epi repair, the number of cutgut chromic sutures, Length and depth of episiotomy. fetal information: fetal weight, fetus position of the occiput posterior or anterior, labor information: duration of the first and second stage, removal of the placenta by hand and etc. You are absolutely right. But as you know, not all interfering factors can be controlled in studies. One of the ways to reduce the effect of these interfering factors is random sampling, which will be considered in this study. However, many of these factors are investigated by the research team. 

Isn’t report the type of random allocation process. Thanks to the accuracy of the reviewer, it was corrected (P8, Par 3).

How will adherence/compliance be assessed? In this study, we will examine the adherence of the samples to drug use and side effects. As mentioned in the method, the use of medicine, painkillers and complications of the intervention will be recorded by the mothers within ten days by means of a designed checklist. Also, during two to three phone calls, in these ten days, they are emphasized about the use of oils. (p11, par 2)

How many times a day were used the oils? Oils will be used 3 times a day (p9, par 2)

Was it used based on individual demand and then will be recorded. After explaining the objectives of the research, any mother who is willing to participate in the study and meets the inclusion criteria will be included. The number of mothers who don’t like to participate in this study, will be recorded.(p9, par 1)

Isn’t report Reliability this research. Reliability of this research added to the method. To ensure the reliability of this research, the person who examines the episiotomy wound on the first and tenth days will be the same for all mothers.(p 8, par 4)

 From the ethical point of view, all three groups should have received painkiller, it should be reported which group used less. Thank you very much for your accuracy, you are absolutely right. This was written in the manuscript. All three groups can use painkillers. Also, people report how many painkillers they used in ten days by self-reporting through a checklist provided to them. This is mentioned in the method. (p 9, par 3)

Report of side effect essential oils? The side effects of essential oils will be assessed and reported. This is mentioned in the method. (p 11, par 2)

when Intensity pain and wound healing will be measured? In the first 24 hours after delivery, and ten days later. This corrected in the method. (p9, the last paragragh) 

How Intensity pain and wound healing will be measured? in REEDA scale for evaluate items; sterile tapes should be used to mark the amount of bruising and redness and... then measure with a ruler. Pain intensity will be measured by VAS based on mother’s self-reporting. About REEDA scale, You are right. But because of the limitations of using sterile tapes in our research, we will use visual inspection.

Limitation? For example, this primiparous they change our location during first days, Especially the tenth day, from the mother's house to her house and they are not present at the previous address. On the 10th day, mothers will be gone to the women's clinic of the hospital for examination. (On the 10th day, gynecologists routinely examine mothers' sutures. Therefore, the address of the mother is not important for us. But one of the major limitations of the current study is that some mothers may not return on the 10th day due to the recovery or because of hospitalization of the baby. This restriction was added to the end of the discussion.

Reviewer #5: 

Thank you very much for giving me the opportunity to review this manuscript.

I congratulate the authors for the work done.

I don't have comment. Thank you very much for your valuable comment.

fereshteh-834

21Aban139322

---

## [Decision Letter · Decision Letter 3]

22 Mar 2024

PONE-D-23-34640R3Randomized Controlled Trial of the Effectiveness of Olive and Black Seed Oil Combination on Pain Intensity and Episiotomy Wound Healing in Primiparous Women: A Study ProtocolPLOS ONE

Dear Dr. Behmanesh,

Thank you for submitting your manuscript to PLOS ONE. After careful consideration, we feel that it has merit but does not fully meet PLOS ONE’s publication criteria as it currently stands. Therefore, we invite you to submit a revised version of the manuscript that addresses the points raised during the review process.

We look forward to receiving your revised manuscript.

Kind regards,

Ahmed Mohamed Maged, MD

Academic Editor

PLOS ONE

Journal Requirements:

**Additional Editor Comments:**

ACADEMIC EDITOR: Please respond to all reviewers comments

Reviewers' comments:

Reviewer's Responses to Questions

**Comments to the Author**

1. Does the manuscript provide a valid rationale for the proposed study, with clearly identified and justified research questions?

Reviewer #4: Yes

2. Is the protocol technically sound and planned in a manner that will lead to a meaningful outcome and allow testing the stated hypotheses?

Reviewer #4: Yes

3. Is the methodology feasible and described in sufficient detail to allow the work to be replicable?

Reviewer #4: Yes

4. Have the authors described where all data underlying the findings will be made available when the study is complete?

Reviewer #4: Yes

5. Is the manuscript presented in an intelligible fashion and written in standard English?

Reviewer #4: Yes

6. Review Comments to the Author

You may also provide optional suggestions and comments to authors that they might find helpful in planning their study.

Reviewer #4: Dear authors

Thank you for your revises. Please correct these two comments as well:

• you record:the number of the painkiller used during 10 days.

• you add in the limitations that to assess wound healing, sterile tape was not used and visual inspection was used.

Sincerely, Reviewer #4

7. PLOS authors have the option to publish the peer review history of their article (what does this mean?). If published, this will include your full peer review and any attached files.

Reviewer #4: No

---

## [Author Response · Author response to Decision Letter 3]

26 Mar 2024

Dear Editor 

PONE-ONE

Thank you for your thorough review and consideration of our manuscript “Randomized controlled trial of the effectiveness of olive and black seed oil combination on pain intensity and episiotomy wound healing in primiparous women: Study protocol" (PONE-D-23-34640R1) that was submitted to the PONE-ONE. Thanks for your kind attention to the manuscript. 

I review the reference list to ensure that it is complete and correct.

Sincerely, Corresponding author

Dr Fereshteh Behmanesh

PhD in Reproductive Health

Reviewers Response

Reviewer #4

Dear authors

Thank you for your revises. Please correct these two comments as well: 

• you record: the number of the painkiller used during 10 days.

 Many thanks for your valuable comments. Corrected. P 9. Par 3

• you add in the limitations that to assess wound healing, sterile tape was not used and visual inspection was used.

 With many respects to you, this sentence added to the limitation. P 14, par 3

---

## [Editor Report · Decision Letter 4]

28 Mar 2024

Randomized Controlled Trial of the Effectiveness of Olive and Black Seed Oil Combination on Pain Intensity and Episiotomy Wound Healing in Primiparous Women: A Study Protocol

PONE-D-23-34640R4

Dear Dr. Behmanesh,

We’re pleased to inform you that your manuscript has been judged scientifically suitable for publication and will be formally accepted for publication once it meets all outstanding technical requirements.

Kind regards,

Ahmed Mohamed Maged, MD

Academic Editor

PLOS ONE
---

## [Editor Report · Acceptance letter]

26 Apr 2024

PONE-D-23-34640R4 

PLOS ONE

Dear Dr. Behmanesh, 

I'm pleased to inform you that your manuscript has been deemed suitable for publication in PLOS ONE. Congratulations! Your manuscript is now being handed over to our production team.

Kind regards, 

on behalf of

Professor Ahmed Mohamed Maged 

Academic Editor

PLOS ONE